# Operator entanglement in local quantum circuits II: solitons in chains of qubits

**Bruno Bertini⋆, Pavel Kos, and Tomaz Prosen**

Department of physics, FMF, University of Ljubljana,
Jadranska 19, SI-1000 Ljubljana, Slovenia

⋆ bruno.bertini@fmf.uni-lj.si

## Abstract

We provide exact results for the dynamics of local-operator entanglement in quantum circuits with two-dimensional wires featuring *ultralocal solitons*, i.e. single-site operators which, up to a phase, are simply shifted by the time evolution. We classify all circuits allowing for ultralocal solitons and show that only dual-unitary circuits can feature moving ultralocal solitons. Then, we rigorously prove that if a circuit has an ultralocal soliton moving to the left (right), the entanglement of local operators initially supported on even (odd) sites saturates to a constant value and its dynamics can be computed exactly. Importantly, this does not bound the growth of complexity in *chiral circuits*, where solitons move only in one direction, say to the left. Indeed, in this case we observe numerically that operators on the odd sublattice have unbounded entanglement. Finally, we present a closed-form expression for the local-operator entanglement entropies in circuits with ultralocal solitons moving in both directions. Our results hold irrespectively of integrability.



# 1   Introduction

This is the second of two papers on the dynamics of local-operator entanglement in local quantum circuits. In the first part of our work [1], which in the following we will refer to as "Paper I", we focussed on the dynamics of operator entanglement in chaotic dual unitary circuits [2]. Here, instead, we do not assume dual-unitarity and we focus on circuits admitting *solitons*: operators that are simply shifted by the time evolution (see below for a precise definition). Specifically, we consider periodically driven spin-$(d-1)/2$ chains with Floquet operator given by

$$\mathbb{U} = \mathbb{U}^{\mathrm{o}}\mathbb{U}^{\mathrm{e}} = \mathbb{T}_{2L}U^{\otimes L}\mathbb{T}_{2L}^{\dagger}U^{\otimes L} = \qquad\qquad . \qquad (1)$$

Following the (diagrammatic) notation of Paper I,

$$U = \qquad\qquad (2)$$

denotes the unitary "local gate" in $\mathrm{End}(\mathbb{C}^d \otimes \mathbb{C}^d)$ and $\mathbb{T}_{\ell}$ the $\ell$−periodic translation by one site. Details of notation exactly follow Paper I. In this framework we call "soliton" a traceless local operator $\tilde{a}$ (in general acting non-trivially on $r$ sites) that, up to a phase, is simply shifted by a number of sites $x$ when conjugated with the Floquet operator (one period of time evolution)

$$\mathbb{U}^{\dagger}\tilde{a}_y\mathbb{U} = e^{i\phi}\tilde{a}_{y+x}, \qquad\qquad \phi \in [0, 2\pi], \qquad\qquad x, y \in \frac{1}{2}\mathbb{Z}_{2L}, \qquad (3)$$

where, for any $b \in \mathrm{End}(\mathbb{C}^{d^r})$, we define $b_y \in \mathrm{End}(\mathbb{C}^{d^L})$ by

$$b_y \equiv \underbrace{\mathbb{1} \otimes \cdots \otimes \mathbb{1}}_{L-1+2y} \otimes b \otimes \underbrace{\mathbb{1} \otimes \cdots \otimes \mathbb{1}}_{L-2y-r-1}. \qquad (4)$$

Due to the strict light-cone structure of the quantum circuit, only a restricted set of values of $x$ are allowed in (3). Specifically $x \in \{-1/2, 0, 1/2, 1\}$ if $y$ is integer and $x \in \{-1, -1/2, 0, 1/2\}$ if $y$ is half-odd-integer. Note that the condition of vanishing trace ensures that $\tilde{a}$ is orthogonal to the identity operator and hence non-trivial.

In this paper we analyse the consequences of the presence of solitons on the dynamics of the local-operator entanglement. For the sake of simplicity, we here consider spin-1/2 chains, i.e. circuits where the local Hilbert space is two-dimensional ($d = 2$), featuring "ultralocal" solitons, i.e. solitons acting non-trivially only on *a single site* ($r = 1$ in Eq. (4)). Specifically, the rest of this paper is laid out as follows. In Section 2 we present a detailed classification of

circuits with ultralocal solitons and two-dimensional local Hilbert space. In Section 3 we recall the definition of the local-operator-entanglement entropies in local quantum circuits and their expression in terms of partition functions on appropriate space-time surfaces. In Section 4 we determine the dynamics of local-operator-entanglement in circuits with solitons presenting closed-form expressions for the entanglement entropies. Finally, Section 5 contains our conclusions. The most technical aspects of our analysis are relegated to the two appendices.

## 2 Classification

Given the definition (3), a natural question is to find all possible quantum circuits, i.e. all possible local gates $U$, admitting solitons. For ultralocal solitons in spin-1/2 chains, this task is explicitly carried out in Appendix A while here we discuss the results. Interestingly, we find that (3) does not have solutions for all allowed values of $x$: only $x = 0$ and $x = 1 - 2\mod(2y, 2) \in \{\pm 1\}$ lead to a consistent solution. In other words solitons can either stay still or move at the maximal speed: if they were initially on integer sites they move to the right and to the left otherwise. The three kinds of allowed solitons impose the following conditions on the local gate

$$\mathbb{U}^\dagger \tilde{a}_y \mathbb{U} = e^{i\phi} \tilde{a}_y \qquad \Longleftrightarrow \qquad \begin{cases} U^\dagger(a \otimes \mathbb{1})U = \pm(b \otimes \mathbb{1}) \\ U^\dagger(\mathbb{1} \otimes b)U = \pm(\mathbb{1} \otimes a) \end{cases}, \qquad (5a)$$

$$\mathbb{U}^\dagger \tilde{a}_y \mathbb{U} = e^{i\phi} \tilde{a}_{y+1} \text{ and } y \in \mathbb{Z}_L \qquad \Longleftrightarrow \qquad U^\dagger(a \otimes \mathbb{1})U = \pm(\mathbb{1} \otimes a), \qquad (5b)$$

$$\mathbb{U}^\dagger \tilde{a}_y \mathbb{U} = e^{i\phi} \tilde{a}_{y-1} \text{ and } y - \frac{1}{2} \in \mathbb{Z}_L \qquad \Longleftrightarrow \qquad U^\dagger(\mathbb{1} \otimes a)U = \pm(a \otimes \mathbb{1}), \qquad (5c)$$

where $a$ and $b$ are some hermitian traceless operators in $\mathrm{End}(\mathbb{C}^2)$ that can be taken Hilbert-Schmidt normalised

$$\frac{1}{2}\mathrm{tr}[aa^\dagger] = 1, \qquad \frac{1}{2}\mathrm{tr}[bb^\dagger] = 1. \qquad (6)$$

The relations (5) highly constrain the dynamics of the circuit. In particular, they generate an exponentially large number of local conservation laws[1] of the form

$$\sum_{y \in \mathbb{Z}} \prod_{x_k} \tilde{a}_{x_k+y}, \qquad (7)$$

where $\{x_k\}$ are either all integer (if (5b)), all half-integer (if (5c)), or mixed (if (5a)). Generically, however, these conservation laws do not follow the Yang-Baxter structure observed in integrable quantum circuits [4]. Indeed, while for integrable circuits the gate can be written as $U = \check{R}(\lambda_0)$ — where $\check{R}(\lambda)$ is a solution of the Yang-Bater equation and $\lambda_0$ a fixed parameter — this is not the case for all gates satisfying (5). This situation is somehow reminiscent to what happens in kinetically constrained models, such as the so called PXP model [5,6]. In both cases there is an ergodicity breaking (known as *scarring* in the latter case) in certain subsectors of the Hilbert space. However, it is currently not clear whether the existence of exponentially many conserved operators is sufficient to enforce an effective reduction of Hilbert space like in the case of PXP model. Other similar cases are soliton gases in reversible cellular automata like the so-called Rule 54 [7–14] or the model of hard core classical particles studied in [15, 16].

In the following subsections we find all possible local gates solving the relations (5a-5c) and use them to determine the constrains imposed by the presence of solitons on the dynamics of ultralocal operators.

---

[1]Note that the expectation values of these "charges" generically oscillate in time with frequency $\phi$. Similar persistent oscillatory quantities have been recently considered in [3].

## 2.1 Still Solitons

Let us start considering circuits with motionless ultralocal solitons, namely with local gates $U$ fulfilling (5a). The first step is to simplify the latter condition. To this aim we note that, since $a$ and $b$ are hermitian, they can be written as

$$a = e^{-i\vec{\alpha}_1\vec{\sigma}}\sigma_3 e^{i\vec{\alpha}_1\vec{\sigma}}, \qquad b = e^{-i\vec{\alpha}_2\vec{\sigma}}\sigma_3 e^{i\vec{\alpha}_2\vec{\sigma}}, \tag{8}$$

where $\vec{\sigma} = (\sigma_1, \sigma_2, \sigma_3)$ is a vector of Pauli matrices and $\vec{\alpha}_1, \vec{\alpha}_2$ vectors of angles. This means that the gauge transformation (see (63))

$$U \mapsto \left(e^{i\vec{\alpha}_1\vec{\sigma}} \otimes e^{i\vec{\alpha}_2\vec{\sigma}}\right) U \left(e^{-i\vec{\alpha}_2\vec{\sigma}} \otimes e^{-i\vec{\alpha}_1\vec{\sigma}}\right), \tag{9}$$

brings (5a) in the following form

$$\begin{cases} U^\dagger(\sigma_3 \otimes \mathbb{1})U = (-1)^{s_1}(\sigma_3 \otimes \mathbb{1}) \\ U^\dagger(\mathbb{1} \otimes \sigma_3)U = (-1)^{s_2}(\mathbb{1} \otimes \sigma_3) \end{cases} \qquad s_1, s_2 \in \{0,1\}. \tag{10}$$

Using the Pauli algebra, it is easy to see that the most general $U \in \mathrm{U}(4)$ solving these relations can be written as (see Appendix B)

$$U = e^{i\phi}((\sigma_1)^{s_1} \otimes (\sigma_1)^{s_2})(e^{-i\eta\sigma_3} \otimes e^{-i\mu\sigma_3}) \cdot e^{-iJ\sigma_3 \otimes \sigma_3} \qquad \phi, \eta, \mu \in [0, 2\pi], \quad J \in [0, \pi/2]. \tag{11}$$

To find the implications of (10) on the dynamics of other ultralocal operators, we consider

$$U^\dagger(\sigma_\beta \otimes \mathbb{1})U, \qquad U^\dagger(\mathbb{1} \otimes \sigma_\beta)U, \qquad \beta \in \{1,2\}. \tag{12}$$

Simple calculations lead to

$$U^\dagger(\sigma_1 \otimes \mathbb{1})U = \cos 2J\,(e^{i2\eta\sigma_3}\sigma_1) \otimes \mathbb{1} - \sin 2J\,(e^{i2\eta\sigma_3}\sigma_2) \otimes \sigma_3 \tag{13a}$$

$$(-1)^{s_1}U^\dagger(\sigma_2 \otimes \mathbb{1})U = \cos 2J\,(e^{i2\eta\sigma_3}\sigma_2) \otimes \mathbb{1} + \sin 2J\,(e^{i2\eta\sigma_3}\sigma_1) \otimes \sigma_3 \tag{13b}$$

$$U^\dagger(\mathbb{1} \otimes \sigma_1)U = \cos 2J\,\mathbb{1} \otimes (e^{i2\mu\sigma_3}\sigma_1) - \sin 2J\,\sigma_3 \otimes (e^{i2\mu\sigma_3}\sigma_2) \tag{13c}$$

$$(-1)^{s_2}U^\dagger(\mathbb{1} \otimes \sigma_2)U = \cos 2J\,\mathbb{1} \otimes (e^{i2\mu\sigma_3}\sigma_2) + \sin 2J\,\sigma_3 \otimes (e^{i2\mu\sigma_3}\sigma_1). \tag{13d}$$

In essence, these relations mean that nothing can move in such a circuit. All ultralocal operators remain at their initial positions and the time evolution causes, at most, a rotation in the Pauli basis and the appearance of still solitons on their sides. An alternative way to pinpoint this "localisation" is to note that the Floquet operator $\mathbb{U}$ built using the gates (11) is essentially the matrix exponential of the classical Ising Hamiltonian. This is just a trivial instance of the "l-bit" Hamiltonian used for the phenomenological modelling of systems in the many-body localised regime [17].

## 2.2 Chiral Solitons

Let us now consider circuits with a single moving soliton, namely local gates $U$ fulfilling either (5b) or (5c). We shall call such solitons *chiral solitons*, as they move in a fixed direction at the speed of light. Since the treatment is very similar in the two cases, we consider only one of them, say the second one. In other words we focus on *left-moving solitons*. As before, we start by simplifying the problem using a gauge transformation. It is easy to see that, with the transformation

$$U \mapsto \left(\mathbb{1} \otimes e^{i\vec{\alpha}_1\vec{\sigma}}\right) U \left(e^{-i\vec{\alpha}_1\vec{\sigma}} \otimes \mathbb{1}\right), \tag{14}$$

the relation (5c) becomes

$$U^\dagger(\mathbb{1} \otimes \sigma_3)U = (-1)^s(\sigma_3 \otimes \mathbb{1}), \qquad s \in \{0,1\}, \tag{15}$$

where we again represented $a$ as in (8). As shown in Appendix B, all possible $U \in U(4)$ fulfilling (15) are parametrised as follows

$$U = e^{i\phi}(u_+ \otimes (\sigma_1)^s e^{-i(\eta_-/2)\sigma_3}) \cdot V[J] \cdot (e^{-i(\mu_-/2)\sigma_3} \otimes v_+), \tag{16}$$

where $\phi, \eta_-, \mu_- \in [0, 2\pi]$, $J \in [0, \pi/2]$, $u_+, v_+ \in SU(2)$ and we introduced

$$V[J] = e^{-i\frac{\pi}{4}\sigma_1 \otimes \sigma_1} e^{-i\frac{\pi}{4}\sigma_2 \otimes \sigma_2} e^{-iJ\sigma_3 \otimes \sigma_3}. \tag{17}$$

Equation (16) is remarkable: it shows that all gates allowing for chiral solitons are *dual-unitary* [2]. Namely, in addition to being unitary they fulfil

$$\sum_{p,q=1,2} \langle \ell\, q|U^\dagger|k\, p\rangle \langle j\, p|U|i\, q\rangle = \delta_{\ell,i}\delta_{k,j}, \qquad \sum_{p,q=1,2} \langle q\, \ell|U^\dagger|p\, k\rangle \langle p\, j|U|q\, i\rangle = \delta_{\ell,i}\delta_{k,j}, \tag{18}$$

where $\{|i\rangle\,; i = 1, 2\}$ is a real, orthonormal basis of $\mathbb{C}^2$.

As before, to find the implications of (15) on the dynamics of other ultralocal operators we consider

$$U^\dagger(\sigma_\alpha \otimes \mathbb{1})U, \qquad U^\dagger(\mathbb{1} \otimes \sigma_\beta)U, \qquad \alpha \in \{1,2,3\}, \ \beta \in \{1,2\}. \tag{19}$$

Let us start with the first group. Expanding in the Pauli basis we have

$$U^\dagger(\sigma_\gamma \otimes \mathbb{1})U = \sum_{\alpha,\beta \in \{0,1,2,3\}} R^\gamma_{\alpha\beta}\ \sigma_\alpha \otimes \sigma_\beta \qquad \gamma \in \{1,2,3\}, \tag{20}$$

where we used the convention $\sigma_0 = \mathbb{1}$. Since $U^\dagger(\sigma_\gamma \otimes \mathbb{1})U$ are hermitian, traceless, and orthonormal we have

$$R^\gamma_{\alpha\beta} \in \mathbb{R}, \qquad R^\gamma_{00} = 0, \qquad \sum_{\alpha,\beta \in \{0,1,2,3\}} R^{\gamma'}_{\alpha\beta}R^{\gamma''}_{\alpha\beta} = \delta_{\gamma'\gamma''}. \tag{21}$$

Therefore (20) in principle contains 14 free real parameters for each fixed $\gamma$. The number of these parameters, however, is reduced by the conditions (15). Indeed, they imply

$$(\sigma_3 \otimes \mathbb{1}) \cdot U^\dagger(\sigma_\gamma \otimes \mathbb{1})U \cdot (\sigma_3 \otimes \mathbb{1}) = U^\dagger(\sigma_\gamma \otimes \mathbb{1})U, \qquad \gamma \in \{1,2,3\}, \tag{22}$$

meaning that the only non-zero coefficients can be

$$\{R^\gamma_{0\beta}, R^\gamma_{3\beta}\}_{\beta,\gamma=1,2,3}. \tag{23}$$

Note that $R^\gamma_{30} = 0$ because of the dual-unitarity property (18). These coefficients can be expressed in terms of the parameters of the local gate as per Eq. (16), however, the final expressions are cumbersome and not particularly instructive, so we decided not to report them. Expanding in the Pauli basis the second group of operators in (19) we have

$$U^\dagger(\mathbb{1} \otimes \sigma_\gamma)U = \sum_{\alpha,\beta \in \{0,1,2,3\}} L^\gamma_{\alpha\beta}\ \sigma_\alpha \otimes \sigma_\beta, \qquad \gamma \in \{1,2\}, \tag{24}$$

where, as above

$$L^\gamma_{\alpha\beta} \in \mathbb{R}, \qquad L^\gamma_{00} = 0, \qquad \sum_{\alpha,\beta \in \{0,1,2,3\}} L^{\gamma'}_{\alpha\beta}L^{\gamma''}_{\alpha\beta} = \delta_{\gamma'\gamma''}. \tag{25}$$

In this case the constraint is

$$(\sigma_3 \otimes \mathbb{1}) \cdot U^\dagger (\mathbb{1} \otimes \sigma_\gamma) U \cdot (\sigma_3 \otimes \mathbb{1}) = -U^\dagger (\mathbb{1} \otimes \sigma_\gamma) U, \qquad \gamma = 1, 2, \qquad (26)$$

implying that the only non-zero coefficients are

$$\{L_{1\alpha}^\gamma, L_{2\alpha}^\gamma\}_{\alpha=0,1,2,3}^{\gamma=1,2}, \qquad (27)$$

once again, even though it is in principle possible, we do not express the coefficients in terms of the parameters of Eq. (16).

The constrains (23) and (27) are relevant for ultralocal operators respectively initialised on integer and half-odd-integer sites. To see this let us consider the schematic representation of the evolution of a ultralocal operator initially on an integer site $y$ (hence following (23)) for a full time step

$$\underset{y}{\circ \bullet \circ \circ} \overset{\Delta t = 1/2}{\longmapsto} \underset{y}{\circ \circ \bullet \circ} + \circ \underset{y}{\textcircled{z}} \bullet \circ \overset{\Delta t = 1/2}{\longmapsto} \underset{y}{\circ \circ \circ \bullet} + \underset{y}{\textcircled{z}} \circ \circ \bullet + \circ \circ \underset{y}{\textcircled{z}} \bullet + \underset{y}{\textcircled{z}} \circ \textcircled{z} \bullet,$$

where $\bullet$ represents a generic ultralocal operator, $\circ$ represents the identity and $\textcircled{z}$ represents $\sigma^z$. In the subsequent evolution the soliton $\textcircled{z}$ will propagate to the left, while $\bullet$ will continue to propagate to the right (both at the maximal speed). In this case the operator $\bullet$ can be thought of as a sort of wave-front moving to the right, which, during each time step, can "shoot" backwards a maximum of two solitons. The dynamic originated by (27), when $y$ is half-odd-integer, is instead completely different. In this case a schematic picture of the evolution looks like

$$\underset{y}{\circ \circ \bullet \circ} \overset{\Delta t = 1/2}{\longmapsto} \underset{y}{\circ \bullet \bullet \circ} \overset{\Delta t = 1/2}{\longmapsto} \underset{y}{\bullet \bullet \circ \bullet} + \underset{y}{\bullet \bullet \textcircled{z} \bullet},$$

therefore the evolution can be understood in terms of a "left moving front" ($\bullet$), which shoots generic operators propagating to the right.

## 2.3 Solitons in Both Directions

Let us now consider local gates fulfilling both the conditions (5b) and (5c). Namely, we consider circuits that have *solitons propagating in both directions*

$$\begin{cases} U^\dagger (\mathbb{1} \otimes a) U = (-1)^{s_2} (a \otimes \mathbb{1}) \\ U^\dagger (b \otimes \mathbb{1}) U = (-1)^{s_1} (\mathbb{1} \otimes b) \end{cases} \qquad s_1, s_2 \in \{0, 1\}. \qquad (28)$$

Representing again $a$ and $b$ as in (8), we have that the gauge transformation

$$U \mapsto \left( e^{i\vec{\alpha}_2 \vec{\sigma}} \otimes e^{i\vec{\alpha}_1 \vec{\sigma}} \right) U \left( e^{-i\vec{\alpha}_1 \vec{\sigma}} \otimes e^{-i\vec{\alpha}_2 \vec{\sigma}} \right), \qquad (29)$$

brings (28) to the form

$$\begin{cases} U^\dagger (\sigma_3 \otimes \mathbb{1}) U = (-1)^{s_1} (\mathbb{1} \otimes \sigma_3) \\ U^\dagger (\mathbb{1} \otimes \sigma_3) U = (-1)^{s_2} (\sigma_3 \otimes \mathbb{1}) \end{cases} \qquad s_1, s_2 \in \{0, 1\}. \qquad (30)$$

As shown in Appendix B, all possible $U \in U(4)$ fulfilling these relations are parametrised as follows

$$U = e^{i\phi} ((\sigma_1)^{s_1} e^{-i(\eta_+/2)\sigma_3} \otimes (\sigma_1)^{s_2} e^{-i(\eta_-/2)\sigma_3}) \cdot V[J] \cdot (e^{-i(\mu_-/2)\sigma_3} \otimes e^{-i(\mu_+/2)\sigma_3}), \qquad (31)$$

where $\phi, \eta_\pm, \mu_\pm, J \in [0, 2\pi]$ and $V[J]$ is defined in (17). Clearly, being a special case of (16), (31) is also *dual-unitary*. Examples of such gates are the dual-unitary parameter line of the trotterized XXZ chain [2, 4, 18]

$$U_{XXZ} = V[J], \tag{32}$$

and the self-dual kicked Ising model [19–21] at the non-interacting point[2]

$$U_{\text{SDKI}} = e^{i\frac{\pi}{4}\sigma_3} \otimes e^{i\frac{\pi}{4}\sigma_3} \cdot V[0]. \tag{33}$$

The form (31) of the gate implies that in the expansions (20) and (24) the only non-vanishing coefficients are

$$\{R_{\alpha\beta}^\gamma\}_{\alpha=0,3;\beta=1,2}^{\gamma=1,2}, \ \{L_{\alpha\beta}^\gamma\}_{\alpha=1,2;\beta=0,3}^{\gamma=1,2}. \tag{34}$$

In particular, expressing them in terms of the parameters of the gate we find

$$U^\dagger(\sigma_1 \otimes \mathbb{1})U = \sin 2J \ \mathbb{1} \otimes (e^{i(\eta_+ + \mu_+)\sigma_3}\sigma_1) + \cos 2J \ \sigma_3 \otimes (e^{i(\eta_+ + \mu_+)\sigma_3}\sigma_2), \tag{35a}$$

$$(-1)^{s_1} U^\dagger(\sigma_2 \otimes \mathbb{1})U = \sin 2J \ \mathbb{1} \otimes (e^{i(\eta_+ + \mu_+)\sigma_3}\sigma_2) - \cos 2J \ \sigma_3 \otimes (e^{i(\eta_+ + \mu_+)\sigma_3}\sigma_1). \tag{35b}$$

Proceeding analogously we have

$$U^\dagger(\mathbb{1} \otimes \sigma_1)U = \sin 2J \ (e^{i(\eta_- + \mu_-)\sigma_3}\sigma_1) \otimes \mathbb{1} + \cos 2J \ (e^{i(\eta_- + \mu_-)\sigma_3}\sigma_2) \otimes \sigma_3, \tag{36a}$$

$$(-1)^{s_2} U^\dagger(\mathbb{1} \otimes \sigma_2)U = \sin 2J \ (e^{i(\eta_- + \mu_-)\sigma_3}\sigma_2) \otimes \mathbb{1} - \cos 2J \ (e^{i(\eta_- + \mu_-)\sigma_3}\sigma_1) \otimes \sigma_3. \tag{36b}$$

As expected, the "front picture" described above for the spreading of operators on integer sites, now holds also for those on half-odd-integer sites, i.e. operators on integer (half-odd-integer) sites create a front moving to the right (left) at the maximal speed and shooting solitons backward.

## 2.4 Coexistent Still and Moving Solitons: No Go

One could in principle envisage circuits featuring coexistent motionless and propagating solitons. Here we show that this is, however, impossible. We proceed by *reductio ad absurdum* and assume that a still and at least one of the moving solitons are both present, specifically

$$U^\dagger(a \otimes \mathbb{1})U = \pm(b \otimes \mathbb{1}), \tag{37}$$

$$U^\dagger(\mathbb{1} \otimes b)U = \pm(\mathbb{1} \otimes a), \tag{38}$$

$$U^\dagger(\mathbb{1} \otimes c)U = \pm(c \otimes \mathbb{1}), \tag{39}$$

with $a, b, c \in \text{End}(\mathbb{C}^2)$ hermitian and traceless. Considering the commutator of $U^\dagger(\mathbb{1} \otimes b)U$ and $U^\dagger(\mathbb{1} \otimes c)U$ and using the last two relations we have

$$U^\dagger(\mathbb{1} \otimes [b, c])U = 0 \quad \Rightarrow \quad [b, c] = 0. \tag{40}$$

Using that $b$ and $c$ are traceless, hermitian operators in $\text{End}(\mathbb{C}^2)$ we find

$$b = \pm c. \tag{41}$$

This, however, implies that (38) and (39) are compatible only if $c = \pm \mathbb{1}$, which contradicts the tracelessness condition.

---

[2]this form of the gate is equivalent to that reported in Ref. [2] up to a gauge transformation.

# 3  Local-Operator Entanglement

The entanglement of a time-evolving operator is defined as the entanglement of the state corresponding to it under a state-to-operator mapping [14, 22, 24–30]. In particular, here we use the "folding mapping" defined in Section 2.1 of Paper I and we consider the entanglement of the (normalised to one) pure state

$$|a_y(t)\rangle, \tag{42}$$

corresponding to the operator $a_y(t)$ via the mapping. In the graphical representation of Paper I this is depicted as

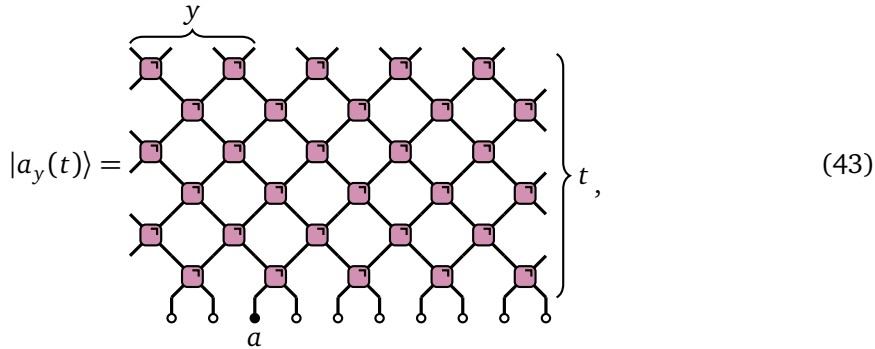

$$\tag{43}$$

where each wire carries a local Hilbert space $\mathbb{C}^2 \otimes \mathbb{C}^2$

$$\Big| = \Big| \Big| \tag{44}$$

and we introduced the "double gate"

$$W = \quad = \quad . \tag{45}$$

Here

$$U^\dagger = \quad , \tag{46}$$

and the upside down red gate means that $U$ is transposed. Finally

$$\Big\downarrow \equiv |\circ\rangle, \qquad \Big\downarrow_a \equiv |a\rangle, \tag{47}$$

where $|\circ\rangle$ corresponds to the identity operator and $|a\rangle$ to a generic ultra-local operator $a$ (both states are normalised to one). Note that in (43) open wires at the sides should be contracted via periodic boundary conditions, while vertical open ends at the top represent a state vector (ket) in $\mathbb{C}^{4^{2L}}$.

From the unitarity of $U$ it follows

$$\quad = \Big| \quad \Big| , \qquad \quad = \Big| \quad \Big| . \tag{48a}$$

$$\quad = \Big) \Big( , \qquad \quad = \Big) \Big( , \tag{48b}$$

where we introduced

$$\text{(diagram)} = W^\dagger. \qquad (49)$$

Using the first of (48a) we see that $|a_y(t)\rangle$ can be simplified out of a lightcone spreading from $y$ at speed 1.

As it is customary, we measure the entanglement of the connected real space region $A = [-L, 0]$ with respect to the rest using the entanglement entropies

$$S^{(n)}(y, t) = \frac{1}{1-n} \log \text{tr}_A\big[\rho_A(t, y; a)^n\big], \qquad n = 1, 2, \ldots, \qquad (50)$$

where we introduced the reduced density matrix

$$\rho_A(t, y; a) = \text{tr}_{\bar{A}}\big[|a_y(t)\rangle\langle a_y(t)|\big] = \qquad . \qquad (51)$$

Here we took $y < t \leq L$. As discussed in Paper I, the entanglement vanishes when the first inequality is violated, while the second inequality is chosen to be in the thermodynamic-limit configuration.

The Renyi entropies (50) are conveniently rewritten in terms of a "corner transfer matrix" [31, 32] $\mathcal{C}[a]$ by noting (see Paper I)

$$\text{tr}_A\big[\rho_A(t, y; a)^n\big] = \text{tr}\big[(\mathcal{C}[a]^\dagger \mathcal{C}[a])^n\big]. \qquad (52)$$

For integer $y$, the matrix elements of the corner transfer matrix read as

$$\langle \alpha_1 \ldots \alpha_{x_-}|\mathcal{C}[a]|\beta_1 \ldots \beta_{x_+}\rangle = \qquad , \qquad \alpha_j, \beta_j \in \{0, \ldots, 3\}, \qquad (53)$$

where $\{|\alpha\rangle, \alpha = 0, \ldots, 3\}$ is the basis of $\mathbb{C}^2 \otimes \mathbb{C}^2$ with

$$|0\rangle = |\circ\rangle, \qquad |1\rangle = |\sigma_1\rangle, \qquad |2\rangle = |\sigma_2\rangle, \qquad |3\rangle = |\sigma_3\rangle, \tag{54}$$

and we introduced the "lightcone" coordinates

$$x_+ \equiv t + y, \qquad x_- \equiv t - y. \tag{55}$$

The hermitian matrices $\mathcal{C}[a]^\dagger \mathcal{C}[a]$ and $\mathcal{C}[a]\mathcal{C}[a]^\dagger$ are represented in terms of the two "row transfer matrices"

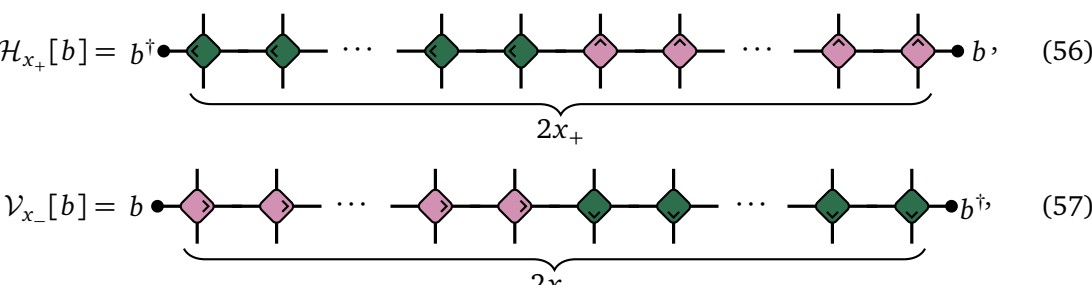

$$\mathcal{H}_{x_+}[b] = \tag{56}$$

$$\mathcal{V}_{x_-}[b] = \tag{57}$$

as follows

$$\langle \alpha_1 \ldots \alpha_{x_+} | \mathcal{C}^\dagger[a]\mathcal{C}[a] | \beta_1 \ldots \beta_{x_+} \rangle = \langle \alpha_1 \ldots \alpha_{x_+}, \beta_{x_+} \ldots \beta_1 | (\mathcal{H}_{x_+}[\mathbb{1}])^{x_-} | a^\dagger \circ \cdots \circ a \rangle, \tag{58}$$

$$\langle \alpha_1 \ldots \alpha_{x_-} | \mathcal{C}[a]\mathcal{C}^\dagger[a] | \beta_1 \ldots \beta_{x_-} \rangle = \langle \alpha_1 \ldots \alpha_{x_-}, \beta_{x_-} \ldots \beta_1 | (\mathcal{V}_{x_-}[\mathbb{1}])^{x_+ -1} \mathcal{V}_{x_-}[a] | \circ \cdots \circ \rangle. \tag{59}$$

The case of $y$ half-odd-integer is recovered from the above formulae by means of the substitutions

$$y \mapsto \frac{1}{2} - y, \qquad \mathcal{H}_x[a] \mapsto \mathcal{V}_x[a], \qquad \mathcal{V}_x[a] \mapsto \mathcal{H}_x[a]. \tag{60}$$

In Paper I we identified a class of completely chaotic dual-unitary circuits by requiring $\mathcal{H}_x[\mathbb{1}]$ and $\mathcal{V}_x[\mathbb{1}]$ to have only $x+1$ eigenvectors of eigenvalue 1. Circuits with solitons are certainly not included in this class, indeed, as discussed in Paper I, the presence of solitons implies that $\mathcal{H}_x[\mathbb{1}]$ and $\mathcal{V}_x[\mathbb{1}]$ have *exponentially many* (in $x$) eigenvectors of with eigenvalue 1.

Note that the corner transfer matrix fulfils

$$\mathrm{tr}[\mathcal{C}[a]^\dagger \mathcal{C}[a]] = 2^{x_+} \langle r_{x_+} | (\mathcal{H}_{x_+}[\mathbb{1}])^{x_-} | a^\dagger \circ \cdots \circ a \rangle = 2^{x_+} \langle r_{x_+} | a^\dagger \circ \cdots \circ a \rangle = 1, \tag{61}$$

where, as explained in Paper I, the "rainbow" state

$$\langle r_x | = \frac{1}{2^x} \sum_{\alpha_1 \alpha_2 \ldots \alpha_x = 0}^{3} \langle \alpha_1 \alpha_2 \ldots \alpha_x \alpha_x \ldots \alpha_2 \alpha_1 | = \underbrace{\overset{\frown}{\ldots \,\text{⌢}\, \ldots}}_{2x}, \tag{62}$$

is a common eigenstate of $\mathcal{V}_x[a]$ and $\mathcal{H}_x[a]$ with eigenvalue 1. Equation (61) is nothing but a statement on the normalisation of the reduced density matrix (cf. (52)). Since $\mathcal{C}[a]^\dagger \mathcal{C}[a]$ is positive semi-definite, the relation (61) implies that its spectrum is contained in $[0, 1]$.

Finally we remark that under the gauge transformation

$$U \mapsto (u \otimes v)U(v^\dagger \otimes u^\dagger), \qquad u, v \in \mathrm{U}(2), \tag{63}$$

the traces of the reduced transfer matrix transform as follows

$$\mathrm{tr}_A[\rho_A(t, y; a)^n] \mapsto \begin{cases} \mathrm{tr}_A[\rho_A(t, y; u^\dagger au)^n] & y \in \mathbb{Z}_L - \frac{1}{2} \\ \mathrm{tr}_A[\rho_A(t, y; v^\dagger av)^n] & y \in \mathbb{Z}_L. \end{cases} \tag{64}$$

This means that the gauge transformation only causes a rotation in the space of ultralocal operators.

# 4 Dynamics of the Operator Entanglement

We are now in a position to study the dynamics of local-operator entanglement in circuits with ultralocal solitons, namely for gates fulfilling (10), (15), or (30) after taking appropriate gauge transformations, e.g. (9). Let us consider the operator entanglement of the normalised operator

$$a = a_1\sigma_1 + a_2\sigma_2 + a_3\sigma_3, \qquad |a_1|^2 + |a_2|^2 + |a_3|^2 = \frac{1}{2}. \qquad (65)$$

## 4.1 Still Solitons

We begin by briefly addressing the trivial case of circuits with motionless solitons, i.e. local gates of the form (11). In this case, after $t$ time steps, (65) reads as

$$a_y(t) = e^{2iJt[\sigma_{3,y-1}\sigma_{3,y}+\sigma_{3,y}\sigma_{3,y+1}]}e^{i(\eta+\mu)t\sigma_{3,y}}\left(a_1\sigma_{1,y} + a_2\sigma_{2,y}\right) + a_3\sigma_{3,y}. \qquad (66)$$

We see that the range of the time evolving operator does not exceed 3, i.e. the operator entanglement is always 0 for $y \notin \{-1/2, 0, 1/2\}$ and its value is bounded by $\log 4$.

## 4.2 Chiral Solitons

Let us now consider the more interesting cases where the circuit features moving ultralocal solitons. As we proved in Sec. 2 (and Appendix A), the latter can not coexist with still solitons. Focussing first on the case of chiral left-moving solitons, we invoke the following property.

**Property 4.1.** *The vector space generated by*

$$\left\{|V_x^{\alpha\beta}\rangle \equiv |\underbrace{\alpha \circ \cdots \circ \beta}_{2x}\rangle, \qquad \alpha, \beta = \{1, 2, 3\}\right\}, \qquad (67)$$

*is closed under the action of $\mathcal{H}_x[\mathbb{1}]$.*

This is easily verified applying the matrix $\mathcal{H}_x[\mathbb{1}]$ (cf. (56)) on the state $|V_x^{\alpha\beta}\rangle$. Indeed, repeated use of (20) with the constrains (23) gives

$$\mathcal{H}_x[\mathbb{1}]|V_x^{\alpha\beta}\rangle = \sum_{\alpha_1,\beta_1=1}^{3} \langle\alpha,\beta|\mathbb{H}|\alpha_1,\beta_1\rangle |V_x^{\alpha_1\beta_1}\rangle, \qquad (68)$$

where we introduced the 9-by-9 matrix $\mathbb{H}$ with elements

$$\langle\alpha,\beta|\mathbb{H}|\gamma,\delta\rangle \equiv \left(R_{0\gamma}^{\alpha}R_{0\delta}^{\beta} + R_{3\gamma}^{\alpha}R_{3\delta}^{\beta}\right). \qquad (69)$$

For integer $y \in \mathbb{Z}_L$ this property has remarkable consequences on the structure of $\mathcal{C}[a]^\dagger\mathcal{C}[a]$: it implies that all the elements of $\mathcal{C}[a]^\dagger\mathcal{C}[a]$ are zero except for a single non-trivial 3-by-3 block, which we denote by $\mathbb{B}[a]$. Namely

$$\langle\alpha_1\ldots\alpha_{x_+}|\mathcal{C}[a]^\dagger\mathcal{C}[a]|\beta_1\ldots\beta_{x_+}\rangle = \langle\alpha_1\ldots\alpha_{x_+}, \beta_{x_+}\ldots\beta_1|(\mathcal{H}_{x_+}[\mathbb{1}])^{x_-}|a^\dagger\circ\cdots\circ a\rangle$$

$$= \left(2\sum_{\alpha,\beta=1}^{3}\langle\alpha_1,\beta_1|(\mathbb{H})^{x_-}|\alpha,\beta\rangle a_\alpha^* a_\beta\right)\prod_{j=2}^{x_+}\delta_{\alpha_j,0}\,\delta_{\beta_j,0}$$

$$\equiv \langle\alpha_1|\mathbb{B}[a]|\beta_1\rangle\prod_{j=2}^{x_+}\delta_{\alpha_j,0}\,\delta_{\beta_j,0}. \qquad (70)$$

Diagonalising the block we have

$$\text{tr}[(\mathcal{C}[a]^\dagger \mathcal{C}[a])^\alpha] = \text{tr}[(\mathbb{B}[a])^\alpha] = e_1^\alpha + e_2^\alpha + e_3^\alpha, \qquad \alpha \in \mathbb{R}, \tag{71}$$

where $e_j \in [0,1]$ are the non-trivial eigenvalues of $\mathcal{C}[a]^\dagger \mathcal{C}[a]$, subject to the constraint (cf. Eq. (61))

$$\text{tr}[\mathcal{C}[a]^\dagger \mathcal{C}[a]] = \text{tr}[\mathbb{B}_a] = e_1 + e_2 + e_3 = 1. \tag{72}$$

This means that, although the eigenvalues depend on $t$ and $y$, they can never all be simultaneously zero and the entanglement is always bounded. In particular we have

$$1 \geq \text{tr}[(\mathcal{C}[a]^\dagger \mathcal{C}[a])^\alpha] \geq 3^{1-\alpha}, \qquad \forall y \in \mathbb{Z}_L, \qquad \forall \alpha \geq 1, \qquad \forall t. \tag{73}$$

This immediately implies

$$S^{(n)}(y,t) \leq \log 3. \tag{74}$$

Repeating the same reasoning for a right-moving soliton we arrive at the following property

**Property 4.2.** *In circuits with two-dimensional local Hilbert space featuring left-moving (right-moving) solitons, the entanglement of ultra-local operators initially supported on integer (half-odd-integer) sites is bounded from above by* $\log 3$.

Importantly, this property constrains only operators on the appropriate sublattice: in circuits with only left-moving solitons the operator entanglement of local operators on half-odd integer sites (and with no overlap with the soliton) is generically unbounded, see Fig. 1 for a representative example. In particular, as demonstrated in Fig. 2 and Tab. 1, the numerical data is consistent with a logarithmic growth. Such a logarithmic growth of local-operator entanglement was observed before in integrable models [23–25, 29] and for the quantum Rule 54 reversible cellular automaton [14], and should be contrasted with the linear growth observed in chaotic circuits [1, 30].

To produce the blue data points in Fig. 1 we explicitly constructed $\mathbb{B}[a]$ by powering the reduced horizontal transfer matrix $\mathbb{H}$. In particular, the limiting value of the entropy can be understood by noting that

$$|\bar{r}_1\rangle \equiv \frac{1}{\sqrt{3}}(2|r_1\rangle - |\circ\circ\rangle) = \frac{1}{\sqrt{3}} \sum_{\alpha=1}^{3} |\alpha\alpha\rangle, \tag{75}$$

is always an eigenstate of $\mathbb{H}$ with eigenvalue 1 (this can be seen directly by using the relations (25)). Assuming that, in the absence of additional symmetries, all other eigenvalues of $\mathbb{H}$ have magnitude strictly smaller than one we have

$$\lim_{t\to\infty} (\mathbb{H})^{x_+} = |\bar{r}_1\rangle\langle\bar{r}_1| \qquad \Rightarrow \qquad \lim_{t\to\infty} \mathbb{B}[a] = \frac{1}{3}\mathbb{1}_3, \tag{76}$$

where $\mathbb{1}_3$ is the 3-by-3 identity. This gives

$$\lim_{t\to\infty} S^{(n)}(y,t) = \log 3, \qquad \forall y \in \mathbb{Z}_L, \qquad \forall n. \tag{77}$$

Table 1: Number of non-zero Schmidt coefficients for local operators on half-odd integer sites in circuits with only left-moving solitons. Note that the number of Schmidt coefficients appears unbounded and the last three entries have values $(t-1)^2$, consistent with a logarithmic growth of $S^{(0)}(0,t) \approx 2\log(t-1)$.

| t | 1 | 2 | 3 | 4 | 5 | 6 | 7 | 8 |
|---|---|---|---|---|---|---|---|---|
| # | 2 | 4 | 8 | 16 | 31 | 49 | 64 | 81 |

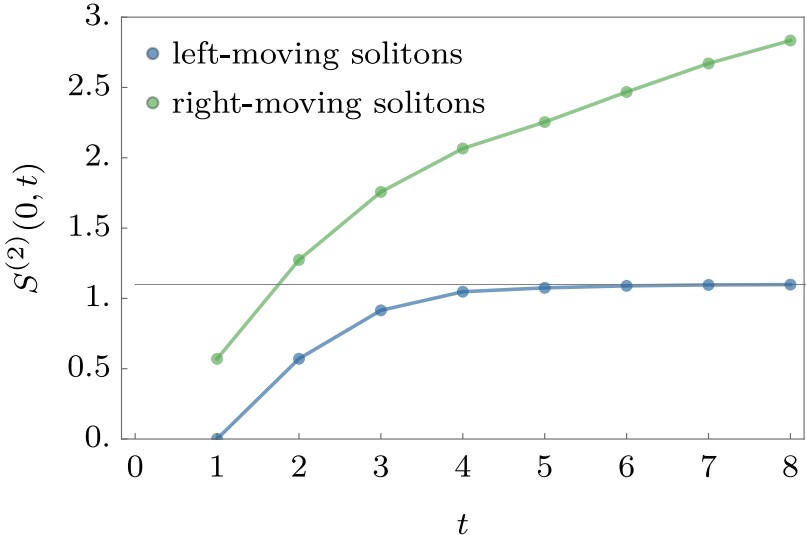

Figure 1: Renyi-2 operator entanglement entropy of $\sigma_1$ at site $y = 0$ versus time for circuits featuring only left-moving (blue) or only right-moving (green) solitons. In the first case the entropy saturates, while in the the second it does not. The limiting value $\log 3$ is depicted with a thin grey line. We used a local gate $U = V[J](\mathbb{1} \otimes v)$ ($U = V[J](v \otimes \mathbb{1})$) for blue (green) data points with $J = -0.3$ and $v$ parametrised by $r = 0.7$ and $\theta = \phi = -0.7$ (see the Supplemental Material of [2] for details on the parametrisation of $v$).

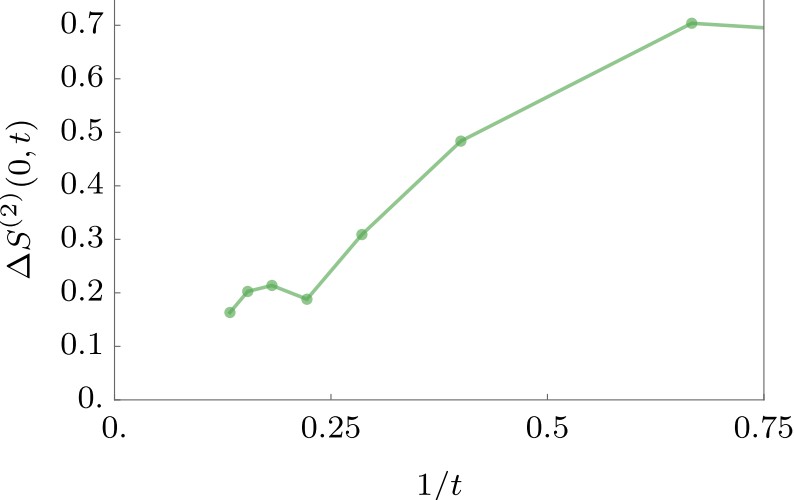

Figure 2: The instantaneous slope of Renyi-2 operator entanglement entropy of $\sigma_1$ at site $y = 0$, $\Delta S^{(2)}(0, t - 1/2) \equiv S^{(2)}(0, t) - S^{(2)}(0, t - 1)$, versus $1/t$ for circuits featuring only right-moving solitons. The decrease in the slope is consistent with a logarithmic growth of $S^{(2)}(0, t)$, which would give $\Delta S^{(2)}(0, t) \sim 1/t$. We used a local gate $U = V[J](v \otimes \mathbb{1})$ with $J = -0.3$ and $v$ parametrised by $r = 0.7$ and $\theta = \phi = -0.7$ (see the Supplemental Material of [2] for details on the parametrisation of $v$).

## 4.3 Solitons in Both Directions

Property 4.2 also implies that in circuits featuring ultralocal solitons moving in *both directions* the operator entanglement of *all local operators is bounded*. We remark that this bound is more stringent than the one recently found in Ref. [14] for the quantum Rule 54 reversible cellular automaton. While there a polynomial growth of the complexity is still possible, in qubits circuits with ultralocal solitons in both directions the complexity is strictly bounded by a constant.

Focussing on this case, we explicitly compute $\mathbb{B}[a]$ as a function of time, initial position, and parameters of the gate (cf. Eqs. (31)) up to similarity transformations. The final expression can be used to determine *all* operator entanglement entropies. Note that this can be in principle done also in the generic case of a single chiral soliton but it produces unwieldy expressions.

For circuits with local gate (31) the reduced horizontal transfer matrix $\mathbb{H}$ reads as

$$\mathbb{H} = \mathbb{R}_3[\theta_+] \otimes \mathbb{R}_3[\theta_+] \cdot \tilde{\mathbb{H}}, \qquad \theta_+ \equiv \eta_+ + \mu_+. \tag{78}$$

Here $\mathbb{R}_3[\theta]$ is a rotation by an angle $\theta \in \mathbb{R}$ around the axis 3 in the three dimensional space

$$\mathbb{R}_3[\theta]\mathbb{R}_3[\theta]^T = \mathbb{R}_3[\theta]\mathbb{R}_3[-\theta] = \mathbb{1}, \tag{79}$$

while $\tilde{\mathbb{H}}$ reads as

$$\tilde{\mathbb{H}} = \sum_{j=1}^{9} \mu_j |\mu_j\rangle \langle \mu_j|, \tag{80}$$

where

$$|\mu_1\rangle = |33\rangle, \qquad |\mu_2\rangle = \frac{1}{\sqrt{2}}(|11\rangle + |22\rangle), \qquad |\mu_3\rangle = \frac{1}{\sqrt{2}}(|12\rangle - |21\rangle), \tag{81}$$

$$|\mu_4\rangle = \frac{1}{\sqrt{2}}(|11\rangle - |22\rangle), \qquad |\mu_5\rangle = \frac{1}{\sqrt{2}}(|12\rangle + |21\rangle), \qquad |\mu_6\rangle = \frac{1}{\sqrt{2}}(|13\rangle - |31\rangle), \tag{82}$$

$$|\mu_7\rangle = \frac{1}{\sqrt{2}}(|23\rangle - |32\rangle), \qquad |\mu_8\rangle = \frac{1}{\sqrt{2}}(|31\rangle + |13\rangle), \qquad |\mu_9\rangle = \frac{1}{\sqrt{2}}(|23\rangle + |32\rangle), \tag{83}$$

and

$$\mu_1 = \mu_2 = (-)^{s_1}\mu_3 = 1, \qquad\qquad \mu_4 = (-)^{s_1}\mu_5 = -\cos(4J)$$
$$(-)^{s_1}\mu_6 = \mu_7 = (-)^{s_1}\mu_8 = \mu_9 = \sin(2J). \tag{84}$$

Importantly

$$\mathbb{R}_3[\theta] \otimes \mathbb{R}_3[\theta] \cdot \tilde{\mathbb{H}} = \tilde{\mathbb{H}} \cdot \mathbb{R}_3[(-)^{s_1}\theta] \otimes \mathbb{R}_3[(-)^{s_1}\theta]. \tag{85}$$

This means that

$$\langle \alpha | \mathbb{B}[a] | \beta \rangle = \langle \alpha, \beta | (\mathbb{R}_3[\xi] \otimes \mathbb{R}_3[\xi]) \cdot \tilde{\mathbb{H}}^{x_+} | a^\dagger a \rangle, \tag{86}$$

where $\xi$ depends on $s_1$ and $x_+$. The matrix $\mathbb{R}_3[\xi] \otimes \mathbb{R}_3[\xi]$, however, does not affect the spectrum of $\mathbb{B}[a]$. Indeed, it only generates a similarity transformation. Explicitly, we have

$$\mathbb{B}[a] = \mathbb{R}_3[\xi]\widetilde{\mathbb{B}}[a]\mathbb{R}_3[\xi]^T, \tag{87}$$

where we set

$$\langle \alpha | \widetilde{\mathbb{B}}[a] | \beta \rangle = \sum_{\alpha_1,\beta_1=1}^{3} 2 \langle \alpha, \beta | \tilde{\mathbb{H}}^{x_+} | \alpha_1, \beta_1 \rangle a^*_{\alpha_1} a_{\beta_1}$$
$$= \sum_{j=1}^{9} \sum_{\alpha_1,\beta_1=1}^{3} 2\mu_j^{x_-} a^*_{\alpha_1} a_{\beta_1} \langle \alpha, \beta | \mu_j \rangle \langle \mu_j | \alpha_1, \beta_1 \rangle. \tag{88}$$

In matrix form it reads as[3]

$$
\begin{aligned}
\widetilde{\mathbb{B}}[a] =& \frac{1}{\sqrt{6}}\lambda_0 + 2\mathrm{Re}[a_1 a_2^*](-\cos(4J))^{x_+}\lambda_1 + 2\mathrm{Im}[a_1 a_2^*]\lambda_2 \\
&+ \left(|a_1|^2 - |a_2|^2\right)(-\cos(4J))^{x_+}\lambda_3 + 2\mathrm{Re}[a_1 a_3^*](\sin(2J))^{x_+}\lambda_4 \\
&+ 2\mathrm{Im}[a_1 a_3^*](\sin(2J))^{x_+}\lambda_5 + 2\mathrm{Re}[a_2 a_3^*](\sin(2J))^{x_+}\lambda_6 \\
&+ 2\mathrm{Im}[a_2 a_3^*](\sin(2J))^{x_+}\lambda_7 + \left(\frac{|a_1|^2 + |a_2|^2 - 2|a_3|^2}{\sqrt{3}}\right)\lambda_8,
\end{aligned}
\tag{89}
$$

where $\{\lambda_j\}$ are the Gell-Mann matrices ($\lambda_0 = \sqrt{\frac{2}{3}}\mathbb{1}_3$). The traces

$$
\mathrm{tr}[(\mathbb{B}[a])^\alpha] = \mathrm{tr}[(\widetilde{\mathbb{B}}[a])^\alpha],
\tag{90}
$$

can then be computed using

$$
\mathrm{tr}[\lambda_a \lambda_b] = 2\delta_{ab},
\tag{91}
$$

and the Gell-Mann algebra

$$
[\lambda_a, \lambda_b] = i f^{abc}\lambda_c,
\tag{92}
$$

where the only non-zero structure constants are

$$
f^{123} = 2, \qquad\qquad f^{147} = f^{246} = f^{257} = f^{345} = 1,
\tag{93}
$$

$$
f^{156} = f^{367} = -1, \qquad\qquad f^{458} = f^{678} = \sqrt{3},
\tag{94}
$$

and all other elements obtained by permutation of the indices ($f^{abc}$ is completely antisymmetric). In particular, we have

$$
\begin{aligned}
S^{(2)}(y,t) = -\log\mathrm{tr}[(\widetilde{\mathbb{B}}[a])^2] =& -\log\Big[\frac{1}{3} + \frac{2}{3}\left(|a_1|^2 + |a_2|^2 - 2|a_3|^2\right)^2 + 8\mathrm{Im}[a_1 a_2^*]^2 \\
&+ 2\left(\left(|a_1|^2 - |a_2|^2\right)^2 + 4\mathrm{Re}[a_1 a_2^*]^2\right)(\cos(4J))^{2x_+} \\
&+ 8|a_3|^2(|a_1|^2 + |a_2|^2)(\sin(2J))^{2x_+}\Big].
\end{aligned}
\tag{95}
$$

Note that $\widetilde{\mathbb{B}}[a]$ and hence the operator-entanglement entropies depend only on the parameter $J$. This means that in all circuits with solitons in both directions the operator entanglement is the same as in, e.g., the dual-unitary trotterised XXZ (cf. (32)). In particular, the free self-dual kicked Ising model (cf. (33)) has $J = 0$ and the operator entanglement is constant.

## 5 Conclusions

We have used the operator entanglement to characterise the complexity of operator dynamics in local quantum circuits on chains of qubits, i.e. quantum circuits with two-dimensional local Hilbert space. Specifically, we have provided a complete classification of circuits possessing *ultralocal solitons* — operators with unit range that are simply translated by the time evolution — and we have shown that the entanglement of a given operator depends on its "initial light-ray", i.e. the line connecting its initial position to the centre of the nearest local gate. For operators with initial light-rays *crossed* by moving solitons the entanglement is bounded from above by $\log 3$ for all times. Instead, in the opposite case the entanglement appears to grow indefinitely as for generic circuits (if the initial operator has no overlap with the conserved

---

[3]we absorbed some factors $(-1)^{s_1 x_+}$ in a redefinition of the Gell-mann matrices that preserves the algebra.

mode). For qubit chains, circuits with solitons (integrable or not) seem to be the only case where the operator complexity does not grow.

Remarkably, we have also proven that the presence of moving solitons immediately implies that the circuit has to be dual-unitary [2] (while the converse is not true: generic dual-unitary circuits do not exhibit solitons and have exponentially growing complexity [1]).

An interesting question for further research is to provide a similar classification for solitons of higher range or for local circuits with larger local Hilbert space. Such circuits can be thought of as toy "coarse grained" versions of integrable models, with the solitons playing the role of quasiparticles, and can be used to explain the generic slow growth of complexity observed in integrable models [14].

## Acknowledgements

We thank Lorenzo Piroli, Marko Medenjak, Vincenzo Alba, Jérôme Dubail, Andrea De Luca, and Tibor Rakovszky for useful discussions. BB thanks LPTMS Orsay for hospitality during the completion of this work.

**Funding Information** This work has been supported by the European Research Council under the Advanced Grant No. 694544 – OMNES, and by the Slovenian Research Agency (ARRS) under the Programme P1-0402.

## A Constrains From Solitons on Local Gates

In this appendix we show how to obtain the conditions (5) from (3). We start by considering (3) and take the scalar product with $\tilde{a}_{y+x}$, which is represented as

$$\text{(diagram)} = e^{i\phi}. \tag{96}$$

Where we have chosen $y$ integer ($y$ half-odd-integer is treated in a totally analogous way) and we followed the notation of Paper I. Depending on $x$ this condition can be written as

$$\frac{1}{2}\text{tr}[\mathcal{M}_+^C(\mathcal{M}_+^C(\tilde{a}))\tilde{a}^\dagger] = e^{i\phi} \qquad x = 1, \tag{97a}$$

$$\frac{1}{2}\text{tr}[\mathcal{M}_+^S(\mathcal{M}_+^C(\tilde{a}))\tilde{a}^\dagger] = e^{i\phi} \qquad x = 1/2, \tag{97b}$$

$$\frac{1}{2}\text{tr}[\mathcal{M}_-^S(\mathcal{M}_+^S(\tilde{a}))\tilde{a}^\dagger] = e^{i\phi} \qquad x = 0, \tag{97c}$$

$$\frac{1}{2}\text{tr}[\mathcal{M}_-^C(\mathcal{M}_+^S(\tilde{a}))\tilde{a}^\dagger] = e^{i\phi} \qquad x = -1/2, \tag{97d}$$

where we introduced the following *unistochastic* single-wire maps (see also Ref. [2])

$$\mathcal{M}_+^C(\tilde{a}) = \text{(diagram)}, \qquad \mathcal{M}_-^C(\tilde{a}) = \text{(diagram)}, \qquad \mathcal{M}_+^S(\tilde{a}) = \text{(diagram)}, \qquad \mathcal{M}_-^S(\tilde{a}) = \text{(diagram)}. \tag{98}$$

Since these maps are all contracting, namely

$$\|\mathcal{M}_\pm^{C/S}(\tilde{a})\|_1 \le \|\tilde{a}\|_1, \tag{99}$$

where $\|A\|_1 = \mathrm{tr}[\sqrt{AA^\dagger}]$ is the trace norm, we can rewrite (97) as

$$\begin{cases} \mathcal{M}_+^C(\tilde{a}) = e^{i\phi_1}\tilde{b} \\ \mathcal{M}_+^C(\tilde{b}) = e^{i(\phi-\phi_1)}\tilde{a} \end{cases} \qquad x = 1, \qquad (100)$$

$$\begin{cases} \mathcal{M}_+^C(\tilde{a}) = e^{i\phi_1}\tilde{b} \\ \mathcal{M}_+^S(\tilde{b}) = e^{i(\phi-\phi_1)}\tilde{a} \end{cases} \qquad x = 1/2, \qquad (101)$$

$$\begin{cases} \mathcal{M}_+^S(\tilde{a}) = e^{i\phi_1}\tilde{b} \\ \mathcal{M}_-^S(\tilde{b}) = e^{i(\phi-\phi_1)}\tilde{a} \end{cases} \qquad x = 0, \qquad (102)$$

$$\begin{cases} \mathcal{M}_+^S(\tilde{a}) = e^{i\phi_1}\tilde{b} \\ \mathcal{M}_-^C(\tilde{b}) = e^{i(\phi-\phi_1)}\tilde{a} \end{cases} \qquad x = -1/2, \qquad (103)$$

where $\tilde{b}$ is a traceless operator in $\mathrm{End}(\mathbb{C}^2)$. Or, equivalently

$$\begin{cases} U^\dagger(\tilde{a} \otimes \mathbb{1})U = e^{i\phi_1}(\mathbb{1} \otimes \tilde{b}) \\ U^\dagger(\tilde{b} \otimes \mathbb{1})U = e^{i(\phi-\phi_1)}\mathbb{1} \otimes \tilde{a} \end{cases} \qquad x = 1, \qquad (104a)$$

$$\begin{cases} U^\dagger(\tilde{a} \otimes \mathbb{1})U = e^{i\phi_1}\mathbb{1} \otimes \tilde{b} \\ U^\dagger(\tilde{b} \otimes \mathbb{1})U = e^{i(\phi-\phi_1)}\tilde{a} \otimes \mathbb{1} \end{cases} \qquad x = 1/2, \qquad (104b)$$

$$\begin{cases} U^\dagger(\tilde{a} \otimes \mathbb{1})U = e^{i\phi_1}\tilde{b} \otimes \mathbb{1} \\ U^\dagger(\mathbb{1} \otimes \tilde{b})U = e^{i(\phi-\phi_1)}\mathbb{1} \otimes \tilde{a} \end{cases} \qquad x = 0, \qquad (104c)$$

$$\begin{cases} U^\dagger(\tilde{a} \otimes \mathbb{1})U = e^{i\phi_1}\tilde{b} \otimes \mathbb{1} \\ U^\dagger(\mathbb{1} \otimes \tilde{b})U = e^{i(\phi-\phi_1)}\tilde{a} \otimes \mathbb{1} \end{cases} \qquad x = -1/2. \qquad (104d)$$

First we note that these relations can be simplified by means of the following lemma.

**Lemma A.1.** *The relation*

$$U^\dagger S^l(\tilde{a} \otimes \mathbb{1})S^l U = e^{i\phi}S^m\tilde{b} \otimes \mathbb{1}S^m, \qquad (105)$$

*with $\tilde{a}, \tilde{b} \in \mathrm{End}(\mathbb{C}^2)$ traceless, is equivalent either to*

$$U^\dagger S^l(a \otimes \mathbb{1})S^l U = +S^m b \otimes \mathbb{1}S^m, \qquad (106)$$

*or to*

$$U^\dagger S^l(a \otimes \mathbb{1})S^l U = -S^m b \otimes \mathbb{1}S^m, \qquad (107)$$

*where $l,m \in \{0,1\}$, $a,b \in \mathrm{End}(\mathbb{C}^2)$ hermitian and traceless and $S$ is the "swap-gate"*

$$S(a \otimes b)S^\dagger = b \otimes a. \qquad (108)$$

*An explicit expression of the swap gate is $S = V[\pi/4]$ (cf. Eq. (17)).*

*Proof.* Let us consider the Hermitian operators

$$h = \tilde{a}\tilde{a}^\dagger - \mathbb{1} \qquad k = \tilde{b}\tilde{b}^\dagger - \mathbb{1}. \qquad (109)$$

It is immediate to see that $h$ and $k$ are traceless and fulfill (106) for $a = h$ and $b = k$. Therefore, if none of $h$ and $k$ is the null operator we can set $a = h$, $b = k$ and the proof is concluded. If, instead, $h = 0$, from (105) follows that also $k = 0$. Therefore the traceless operators $\tilde{a}$ and $\tilde{b}$ are also unitary. This means

$$\tilde{a} = e^{i\theta_1}u\sigma_3 u^\dagger, \qquad \tilde{b} = e^{i\theta_2}v\sigma_3 v^\dagger, \qquad (110)$$

where $\theta_1, \theta_2 \in \mathbb{R}$ and $u, v \in \mathrm{SU}(2)$. This means that the hermitian operators $a = e^{-i\theta_1}\tilde{a}$ and $b = e^{-i\theta_2}\tilde{b}$ fulfil (105) with $\phi \to \phi - \theta_1 + \theta_2$. However, since $a$ and $b$ are hermitian we have

$$e^{i(\phi-\theta_1+\theta_2)}S^k(\mathbb{1} \otimes b)S^k = U^\dagger S^h(a \otimes \mathbb{1})S^h U = \left(U^\dagger S^h(a \otimes \mathbb{1})S^h U\right)^\dagger =$$
$$= e^{-i(\phi-\theta_1+\theta_2)}S^k(\mathbb{1} \otimes b)S^k, \tag{111}$$

meaning that $e^{i\phi-\theta_1+\theta_2} = \pm 1$. This concludes the proof. $\qquad\square$

Using Lemma A.1 we can rewrite (104) as follows

$$\begin{cases} U^\dagger(a \otimes \mathbb{1})U = \pm\mathbb{1} \otimes b \\ U^\dagger(b \otimes \mathbb{1})U = \pm\mathbb{1} \otimes a \end{cases} \qquad x = 1, \tag{112a}$$

$$\begin{cases} U^\dagger(a \otimes \mathbb{1})U = \pm\mathbb{1} \otimes b \\ U^\dagger(b \otimes \mathbb{1})U = \pm a \otimes \mathbb{1} \end{cases} \qquad x = 1/2, \tag{112b}$$

$$\begin{cases} U^\dagger(a \otimes \mathbb{1})U = \pm b \otimes \mathbb{1} \\ U^\dagger(\mathbb{1} \otimes b)U = \pm\mathbb{1} \otimes a \end{cases} \qquad x = 0, \tag{112c}$$

$$\begin{cases} U^\dagger(a \otimes \mathbb{1})U = \pm b \otimes \mathbb{1} \\ U^\dagger(\mathbb{1} \otimes b)U = \pm a \otimes \mathbb{1} \end{cases} \qquad x = -1/2. \tag{112d}$$

Let us now show that (112b) and (112d) cannot be fulfilled. Since the reasoning is very similar in the two cases we consider only (112b). We proceed by *reductio ad absurdum*. Assuming the two conditions (112b), we take the commutator of the two and find

$$U^\dagger([a, b] \otimes \mathbb{1})U = 0, \tag{113}$$

meaning

$$[a, b] = 0. \tag{114}$$

Since $a, b \in \mathrm{End}(\mathbb{C}^2)$ are Hermitian and traceless we conclude

$$b = \pm a. \tag{115}$$

This implies

$$a \otimes \mathbb{1} = \pm\mathbb{1} \otimes a, \tag{116}$$

which is possible only if $a = \mathbb{1}$, which is not traceless. This leads to a contradiction.

Let us now consider (112a) and show that the two conditions are equivalent to

$$U^\dagger(a \otimes \mathbb{1})U = \pm\mathbb{1} \otimes a, \tag{117}$$

with $a$ hermitian and traceless. To prove this we distinguish two cases

(i) $[a, b] = 0$

(ii) $[a, b] \neq 0$.

In the first case we have $a = \pm b$ so that (117) is fulfilled. In the second case we see that (117) is fulfilled by $[a, b]$. This concludes our analysis.

# B   Classification of all Local Gates with Solitons

Here we prove the following property

**Property B.1.** *The most general gate $U \in U(4)$ fulfilling*

$$U^\dagger(\sigma_3 \otimes \mathbb{1})U = (-1)^s(\sigma_3 \otimes \mathbb{1}), \qquad s \in \{0,1\} \tag{118}$$

*is given by*

$$U = e^{i\phi}((\sigma_1)^s e^{-i(\eta_-/2)\sigma_3} \otimes u_+) \cdot e^{-iJ\sigma_3 \otimes \sigma_3} \cdot (e^{-i(\mu_-/2)\sigma_3} \otimes v_+), \tag{119}$$

*where $\phi, \eta_-, \mu_- \in [0, 2\pi]$, $J \in [0, \pi/2]$ and $u_+, v_+ \in SU(2)$.*

Note that, solving this problem we also find the most general gate fulfilling

$$W^\dagger(\mathbb{1} \otimes \sigma_3)W = (-1)^s(\sigma_3 \otimes \mathbb{1}), \tag{120}$$

by posing $W = SU$ where $S = V[\pi/4]$ is the swap-gate (108). This gives

$$W = e^{i\phi}(u_+ \otimes (\sigma_1)^s e^{-i(\eta_-/2)\sigma_3}) \cdot V[J - \pi/4] \cdot (e^{-i(\mu_-/2)\sigma_3} \otimes v_+). \tag{121}$$

In the proof we make often use of the following Lemma

**Lemma B.1.** *The most general matrix $u \in SU(2)$ such that*

$$\mathrm{tr}[\sigma_3 u \sigma_3 u^\dagger] = 2(-1)^s \tag{122}$$

*can be written as*

$$u = (\sigma_1)^s e^{i\theta\sigma_3}, \qquad \theta \in [0, 2\pi]. \tag{123}$$

The proof is immediately obtained by representing $\sigma_1^s u$ as $e^{i\vec{\alpha}\cdot\vec{\sigma}}$: for the sake of brevity we omit the details.

To prove property B.1 we consider a generic matrix in $U \in U(4)$, which, according to Refs. [33, 34], is written as

$$U = e^{i\phi}(u_- \otimes u_+)V[J_1, J_2, J_3](v_- \otimes v_+), \tag{124}$$

where $\phi \in [0, 2\pi]$, $J_1, J_2, J_3 \in [0, \pi/2]$, $u_\pm, v_\pm \in SU(2)$ and we defined

$$V[J_1, J_2, J_3] = \exp[-i(J_1\sigma_1 \otimes \sigma_1 + J_2\sigma_2 \otimes \sigma_2 + J_3\sigma_3 \otimes \sigma_3)]. \tag{125}$$

Plugging (124) into (118) we have

$$V[J_1, J_2, J_3] = (-1)^s(u_-^\dagger\sigma_3 u_- \otimes \mathbb{1})V[J_1, J_2, J_3](v_-\sigma_3 v_-^\dagger \otimes \mathbb{1}) \tag{126}$$

and we immediately see that (118) does not impose any constraint on the matrices $u_+, v_+ \in SU(2)$ and on the phase $\phi$. Then, using that

$$V[J_1, J_2, J_3] = \sum_{\beta=0}^3 V_\beta(J_1, J_2, J_3)\sigma_\beta \otimes \sigma_\beta, \tag{127}$$

with

$$V_0(J_1, J_2, J_3) = \cos(J_1)\cos(J_2)\cos(J_3) - i\sin(J_1)\sin(J_2)\sin(J_3), \tag{128}$$

$$V_\beta(J_1, J_2, J_3) = \cos(J_\beta)\prod_{\alpha\neq\beta}\sin(J_\alpha) - i\sin(J_\beta)\prod_{\alpha\neq\beta}\cos(J_\alpha), \qquad \beta \in \{1, 2, 3\}, \tag{129}$$

we express the condition (126) in components (in the basis $\{\sigma_\alpha \otimes \sigma_\beta\}$ ) as follows

$$V_\beta(J_1, J_2, J_3)\mathrm{tr}[u_-^\dagger \sigma_3 u_- \sigma_\beta v_- \sigma_3 v_-^\dagger \sigma_\alpha] = 2(-1)^s \delta_{\alpha,\beta} V_\beta(J_1, J_2, J_3). \tag{130}$$

We now show that if all $V_\beta(J_1, J_2, J_3)$ are non-zero these conditions cannot be all simultaneously satisfied. To prove it we start considering the case $\alpha = \beta = 0$. Since all the coefficients are non zero we have

$$\mathrm{tr}[\sigma_3 u_- v_- \sigma_3 v_-^\dagger u_-^\dagger] = 2(-1)^s, \tag{131}$$

which, using Lemma B.1, implies

$$u_- v_- = (\sigma_1)^s e^{i\theta\sigma_3}. \tag{132}$$

This also solves all conditions (130) where one among $\alpha$ and $\beta$ is 0. Considering now $\alpha = \beta = 3$ we have

$$\mathrm{tr}[\sigma_3 u_- \sigma_3 u_-^\dagger \sigma_3 u_- \sigma_3 u_-^\dagger] = 2 \tag{133}$$

implying

$$u_- \sigma_3 u_-^\dagger = \pm\sigma_3. \tag{134}$$

Considering then $\alpha = \beta = 1$ we have

$$-2 = \mathrm{tr}[\sigma_3 \sigma_1 \sigma_3 \sigma_1] = 2, \tag{135}$$

which is a contradiction. Therefore, at least one of the coefficients has to vanish. We distinguish two cases

(i) $V_0(J_1, J_2, J_3) = 0$

(ii) $V_{\bar\alpha}(J_1, J_2, J_3) = 0, \qquad \bar\alpha \in \{1, 2, 3\}$.

Let us start from the case (i): in this case we should have $J_{\bar\alpha} = 0$ and $J_{\bar\beta} = \pi/2$ for $\bar\alpha, \bar\beta \in \{1, 2, 3\}$. This means that the only non trivial relations (130) are

$$\cos(J_\gamma)\mathrm{tr}[u_-^\dagger \sigma_3 u_- \sigma_{\bar\beta} v_- \sigma_3 v_-^\dagger \sigma_\alpha] = 2(-1)^s \delta_{\alpha,\bar\beta} \cos(J_\gamma), \tag{136}$$

$$\sin(J_\gamma)\mathrm{tr}[u_-^\dagger \sigma_3 u_- \sigma_{\bar\alpha} v_- \sigma_3 v_-^\dagger \sigma_\alpha] = 2(-1)^s \delta_{\alpha,\bar\alpha} \sin(J_\gamma), \tag{137}$$

with $\gamma \neq \bar\alpha, \bar\beta$. First we note that, if $J_\gamma = 0$ the relations simplify to

$$\mathrm{tr}[u_-^\dagger \sigma_3 u_- \sigma_{\bar\beta} v_- \sigma_3 v_-^\dagger \sigma_\alpha] = 2(-1)^s \delta_{\alpha,\bar\beta}. \tag{138}$$

Considering $\alpha = \bar\beta$ using the Lemma B.1 we then conclude

$$u_- \sigma_{\bar\beta} v_- = (\sigma_1)^s e^{i\theta\sigma_3}. \tag{139}$$

This also fulfils all other relations (138). Plugging back into (124) we find

$$U = i e^{i\phi}(\sigma_1)^s e^{i\theta\sigma_3} \otimes u_- \sigma_\beta v_-, \tag{140}$$

which is of the form (119). Analogous considerations hold for $J_\gamma = \pi/2$. If $J_\gamma \neq 0, \pi/2$ the first relations are still solved by (139), while the second become

$$\mathrm{tr}[\sigma_3 u_- \sigma_{\bar\alpha} \sigma_{\bar\beta} u_-^\dagger \sigma_3 u_- \sigma_\alpha \sigma_{\bar\beta} u_-^\dagger] = 2\delta_{\alpha,\bar\alpha}. \tag{141}$$

Using Lemma B.1 we then conclude

$$u_- \sigma_{\bar\alpha} \sigma_{\bar\beta} u_-^\dagger = \pm i u_- \sigma_\gamma u_-^\dagger = e^{i\theta\sigma_3}, \tag{142}$$

which implies

$$u_-\sigma_\gamma u_-^\dagger = \pm\sigma_3. \tag{143}$$

Plugging back into (124) again produces a gate of the form (119). Let us now move to the case (ii): this case implies $(J_{\bar\alpha}, J_{\bar\beta}) = (0,0), (\pi/2, \pi/2)$ for $\bar\alpha, \bar\beta \in \{1,2,3\}$. We consider the first option, as the second can be recovered from the first one by noting

$$V[\pi/2, \pi/2, J_3] = -iV[0,0,\pi/2 + J_3]. \tag{144}$$

If $J_{\bar\alpha} = J_{\bar\beta} = 0$ the conditions (130) become

$$\cos(J_\gamma)\mathrm{tr}[u_-^\dagger\sigma_3 u_- v_-\sigma_3 v_-^\dagger\sigma_\alpha] = 2(-1)^s\delta_{\alpha,0}\cos(J_\gamma), \tag{145}$$

$$\sin(J_\gamma)\mathrm{tr}[u_-^\dagger\sigma_3 u_-\sigma_\gamma v_-\sigma_3 v_-^\dagger\sigma_\alpha] = 2(-1)^s\delta_{\alpha,\gamma}\sin(J_\gamma), \qquad \gamma \neq \bar\alpha, \bar\beta. \tag{146}$$

If $J_\gamma = 0$ the only remaining relation is the first and we have

$$u_- v_- = (\sigma_1)^s e^{i\theta\sigma_3}, \tag{147}$$

which, plugging back into (124) gives a gate of the form (119). An analogous reasoning applies for $J_\gamma = \pi/2$. Finally, if $J_\gamma \neq 0, \pi/2$ we have that (145) are still solved by (147) while the second relations give

$$u_-\sigma_\gamma u_-^\dagger = \pm\sigma_3. \tag{148}$$

This again produces a gate of the form (119) and concludes the proof. $\qquad\square$

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
