# Peer review of "Operator Entanglement in Local Quantum Circuits II: Solitons in Chains of Qubits"

_SciPost Physics, doi:SciPost Phys. 8, 068 (2020)_

## Round 1 · Referee Report · Jerome Dubail (Referee 1) · 2019-11-8

Strengths

1- Very interesting problem, methods and conclusions

2- Provides exhaustive classification of unitary circuit models with solitons (with local Hilbert space dimension d=2)

3- Innovative tools, technically very impressive

Weaknesses

1- The conclusions in the case with solitons propagating only in one direction could be perhaps be expanded and clarified

Report

Like its companion paper, this is an excellent manuscript which deals with the problem of the growth of operator entanglement of local operators in Heisenberg picture (or 'local operator entanglement'). Paper I was dealing with the chaotic case; this Paper II deals with circuit models that sustain stable quasi-particles or solitons and therefore are expected to behave differently. The problems solved here and the methods used are slightly different from those in Paper I.

An important part of this Paper II deals with the classification of circuit models that sustain solitons (section 2 and appendices). This analysis is quite technical but it is very useful, as it clearly singles out three classes of unitary circuits with peculiar dynamics. Those models could turn out to be useful to investigate plenty of other interesting physics questions, not just the problem of local operator entanglement which motivates this paper.

The manuscript is very well written, although I think a few minor points could be clarified (see list of questions/remarks below). One more important remark is that the most peculiar behavior is found for the class of circuit models that sustain solitons going in one direction only, and this is insufficiently emphasized. I believe the physical discussion of that class of circuits could be expanded, and their phenomenology should be explained more clearly. Also, the authors cautiously conclude that the local operator entanglement grows unbounded in that case (for some subset of operators); however it is unclear whether they think that growth is linear as in Paper I, or if it is sublinear (maybe logarithmic as in Ref. [14]). I understand that, for now, this may be out-of-reach analytically, but the authors also did some numerics (as displayed in Fig. 1) so it would be interesting to have a discussion of their conclusions/conjectures about this peculiar case.

Otherwise I am happy to recommend publication of this impressive work in Scipost.

Requested changes

1- I think the observation that some operators have unbounded growth in the 'chiral case' (i.e. circuits with solitons moving only in one direction) is important, and it should appear in the abstract.

2- In the definition of a 'soliton' in the introduction, the meaning of 'ultra-local' should perhaps be recalled. Also, why is a 'soliton' restricted to live on a single site? In general, can't one imagine a more extended object, created by an operator acting on a few neighboring sites, and still satisfying Eq. (3)?

3- In Eqs. (5b) and (5c) it would probably be clearer to write 'and' rather than put a wedge

4- The discussion below Eq. (7) about Yang-Baxter integrability could be clarified. It is unclear what 'Yang-Baxter' means in the context of unitary circuits, because Yang-Baxter is an equation that involves spectral parameters and there are no such parameters in the circuit models; so the authors need to explain what they mean here. The next sentence about 'this somehow being reminiscent of kinetically constrained models' and of 'scarring' also sounds mysterious to me; perhaps the authors can explain this better.

4- Eq. (11): what are $s$ and $s'$?

5- In the next sentence, 'To find the implications of (15)', the ref. '(15)' is probably wrong, I guess it should be '(10)'. A similar problem seems to have occured several times in the manuscript, with incorrect references to equations, please check this.

6- Short discussion after Eqs. (13): 'nothing can move in such a circuit'. This is similar to a localized phase; then is this an interesting model to study localization? Maybe the authors could quickly comment on that.

7- The discussion in the last 5 lines of Sec. 2.2 about 'operators on integer sites generating a front moving opposite to the solitons' and 'fronts shooting back solitons and other generic operators' is not very clear. Since this class of circuit models plays an important role later in the paper, I think that paragraph could be expanded quite a lot. Perhaps a figure could also make that discussion clearer.

8- In Eq. (47), how are the two states $\left| \circ \right>$ and $\left| a \right>$ normalized?

9- for notational consistency the Renyi index '$\alpha$' in Eqs. (72)-(73) should probably be replaced by '$n$'

10- The paragraph after 'Property 4.2', which mentions the unbounded growth of operators on half-integer sites, could be expanded. This is an important result of this paper, so it should be discussed more. In particular, do the authors think the growth is linear? or sublinear? maybe logarithmic? This does not look obvious from Fig. 1.

11- The conclusion of Sec. 4.3 would probably be clearer if the final result (95) was given directly for the Renyi entropy (instead of the present form of Eq. (95), where the reader needs to recall what ${\rm tr}[(\tilde{\mathbb{B}}[a])^2]$ is).

12- In the conclusion, I don't quite understand the sentence 'for qubit chains, circuits with solitons (integrable or not) seem to be the only case where the operator complexity does not grow'. What about qubit chains that map to free fermions (XX chain, Ising chain, etc.)? Those have Hamiltonian evolution and the operator complexity is also known to remain constant for a subset of operators (see e.g. Ref. [22]). In what sense is that different, in terms of operator complexity?

13- some typos:
page 2: 'In Section 3 recall' -> 'In Section 3 we recall'
page 13: Eq. (83) is an empty line
page 14: 'originates a similarity' -> 'originates from a similarity'

---

## Round 1 · Referee Report · Anonymous (Referee 5) · 2019-12-11

Report

Referee report: Operator entanglement in local quantum circuits II..., by Bertini, Kos, and Prosen

This is a thorough and carefully-written study of the dynamics of a class of spin 1/2 Floquet models featuring “solitons”.

In the context of this work a “soliton” is defined as a single-site operator whose Heisenberg evolution amounts essentially to a translation. The authors classify such circuits and show that the existence of solitons restricts the dynamics of other operators, that are not solitons, in an interesting way.

The authors have intentionally limited their scope to a particular class of Floquet models for spin 1/2. These models can be represented as quantum circuits built from a single two-site unitary. This limitation means it is not obvious to what extent these results generalize, for example to models with non-nearest-neighbor interactions or spin greater than 1/2. For example, it is evident that the no-go theorem on having solitons propagating in both directions does not hold if the number of states per site is increased to four. However, the advantage of this restriction is that the authors have been able to obtain very complete and explicit results for a set of models that is still quite rich.

This is a useful contribution in view of the precision and clarity of the results, and I recommend publication in SciPost.

Other comments:

Definition of soliton: the authors’ definition of a soliton is stated in the introduction but the choice of definition is not motivated. The authors may wish to consider whether it would be useful to the reader to explain why this definition is natural (even if just by making the simple point that such solitons have an implication for the dynamics of states). For example, can they draw an analogy with structures in better-known integrable systems?

Is it possible to justify the restriction to single-site operators? For the set of models studied, are solitons of larger size (that do not reduce trivially to groups of smaller ones) ruled out?

Equation 7: notation could be clarified
Equation 11: s, s’ -> s_1, s_2

The diagrammatic notation for tensors in Sec 3 relies on colors and is confusing in a black and white printout.

The rainbow state that appears here is an example of a “permutation state” of the kind that appears in random circuits (e.g.
https://arxiv.org/abs/1907.09581,https://arxiv.org/abs/1804.09737) written in a different convention for folding the circuit.

The authors mention that in the models with a single chirality of soliton some operators can have unbounded entanglement growth. Can the authors comment on the functional form of this growth? Do they believe it is sublinear in time?

In the same models, the authors show in Section 2 that the other type of non-soliton operator has a front that advances while shooting out solitons backwards. Later the authors show by a separate calculation using the transfer matrix formalism that such operators have bounded entanglement. Is there any simple explanation for this result using the direct picture of Section 2 for the terms that can appear in the expansion of the operator?

In the conclusions the authors mention that the circuits they study can be viewed as toy coarse-grained models for integrable systems. Can this analogy be elucidated a little more? For example, do the models with solitons in only one direction have any analogy with integrable systems?

---

## Round 2 · Referee Report · Anonymous (Referee 2) · 2020-2-1

Report

After the authors' changes this interesting paper is now ready for publication.

---

## Round 2 · Referee Report · Jerome Dubail (Referee 1) · 2020-2-5

Report

The authors have answered all my questions/comments and have elaborated on the points that needed clarification. I am happy to recommend this very nice work for publication in Scipost.

---

## Round 2 · Referee Report · Anonymous (Referee 3) · 2020-2-10

Strengths

1: Important follow-up on Paper I " Maximally Chaotic Dual-Unitary Circuits" , considering now cases where explicit local conserved operators are identified.

2: Precise proofs and statements, full classification of ultra-local soliton-compatible operators.

3: Very interesting results of the contrasting behavior of entanglement of operators depending on the chirality of solitons and the even/odd site bearing the original operator;

4: Precise quantitative statements regarding the behavior of entanglement when two chiralities are present.

Weaknesses

1: Purely technical: can only be published once Paper I is finally accepted and published, which may delay publication.

2: Technical question: Explicit comparison between Paper I and Paper II regarding the behavior of eigenvalues/eigenvectors of dual transfer matrices is lacking.

Report

This paper offers a well-written, precise and very interesting pendant to Paper I, by the same authors, submitted to the same journal. It considers sets of dynamical quantum circuits admitting explicit local conserved operators. The results are very interesting, both analytical and numerical; clearly stated and proved, which makes this paper an indispensable and contrasting follow-up to Paper I.

The possibility of joining both papers may be considered, since the subject matter is the same and so is the starting point ( quantum circuits
dynamics ) but it would yield a long ( 40+ pages) manuscript and I will not insist on it.

Comparison with the cases treated in I would benefit from a comparison between results of section 4.2 Paper II and the "maximally chaotic property" such as is characterized in I (Def. 4.1) as a restriction on eigenvectors/eigenvalues of the dual transfer matrices. The discrepancy between the two cases clearly occurs when deriving formulae (70)-(73) of paper II and it may be useful to point out explicitely where the difference lies regarding eigenvalues.

Recommanded for publication after some issues are clarified.

Requested changes

1: A precise comparison between the eigenvalues/eigenvector properties of transfer matrices yielding respectively linear behavior in I and log or constant behavior in II.

---

## Round 2 · Author Response

We thank Dr Dubail for his thorough reading of our paper, for his positive assessment, and for his relevant comments and suggestions. In the following we address his comments in a point-by-point fashion starting from his main remark.

Referee: "One more important remark is that the most peculiar behavior is found for the class of circuit models that sustain solitons going in one direction only, and this is insufficiently emphasized. I believe the physical discussion of that class of circuits could be expanded, and their phenomenology should be explained more clearly. Also, the authors cautiously conclude that the local operator entanglement grows unbounded in that case (for some subset of operators); however it is unclear whether they think that growth is linear as in Paper I, or if it is sublinear (maybe logarithmic as in Ref. [14]). I understand that, for now, this may be out-of-reach analytically, but the authors also did some numerics (as displayed in Fig. 1) so it would be interesting to have a discussion of their conclusions/conjectures about this peculiar case."

To address this point we expanded the discussion of the circuits with chiral solitons. Furthermore, we included an additional figure (Fig. 2) and additional table (Tab. 1), which show that the numerical results are consistent with the logarithmic growth of the operator entanglement entropy for the case where the growth is not bounded.

Referee: "I think the observation that some operators have unbounded growth in the 'chiral case' (i.e. circuits with solitons moving only in one direction) is important, and it should appear in the abstract."

Thanks for the suggestion: we agree. This point is now mentioned in the revised abstract.

Referee: "In the definition of a 'soliton' in the introduction, the meaning of 'ultra-local' should perhaps be recalled. Also, why is a 'soliton' restricted to live on a single site? In general, can't one imagine a more extended object, created by an operator acting on a few neighboring sites, and still satisfying Eq. (3)?"

It is surely possible con consider circuits with solitons with larger support. Providing a complete analysis of the general case, however, goes beyond the scope of the present paper (it is somewhat similar to considering ultralocal solitons in circuits with higher Hilbert space). Here we aim at studying the simplest possible case: ultralocal solitons in circuits of qbits.

To reflect this, in the revised version we define the solitons in general (for generic local quantum circuits and soliton range) while we clearly state that we will only analyse the case of ultralocal solitons on circuits of qbits.

Referee: "In Eqs. (5b) and (5c) it would probably be clearer to write 'and' rather than put a wedge"

Changed.

Referee: "The discussion below Eq. (7) about Yang-Baxter integrability could be clarified. It is unclear what 'Yang-Baxter' means in the context of unitary circuits, because Yang-Baxter is an equation that involves spectral parameters and there are no such parameters in the circuit models; so the authors need to explain what they mean here. The next sentence about 'this somehow being reminiscent of kinetically constrained models' and of 'scarring' also sounds mysterious to me; perhaps the authors can explain this better."

Integrable circuits have the local gate given by an integrable R-matrix (fulfilling the Yang-Baxter equation) at a specific spectral point. This is explained more in detail in the revised version where we also clarified the subsequent sentence.

Referee: "Eq. (11): what are s and s′"

We thank the referee for spotting this typo: They are s1 and s2 from the previous equation.

Referee: "In the next sentence, 'To find the implications of (15)', the ref. '(15)' is probably wrong, I guess it should be '(10)'. A similar problem seems to have occured several times in the manuscript, with incorrect references to equations, please check this."

Once again, we thank the referee for spotting error in referencing (we carefully checked all other equation referencing in the paper and did not find any other mistake).

Referee: "Short discussion after Eqs. (13): 'nothing can move in such a circuit'. This is similar to a localized phase; then is this an interesting model to study localization? Maybe the authors could quickly comment on that."

We expanded the discussion in the revised version.

Referee: "The discussion in the last 5 lines of Sec. 2.2 about 'operators on integer sites generating a front moving opposite to the solitons' and 'fronts shooting back solitons and other generic operators' is not very clear. Since this class of circuit models plays an important role later in the paper, I think that paragraph could be expanded quite a lot. Perhaps a figure could also make that discussion clearer."

We expanded the discussion and added two pictorial equations, we hope that the point is now clearer.

Referee: "In Eq. (47), how are the two states |∘⟩ and |a⟩ normalized?"

To one. We added this explanation in the parenthesis after Eq. 47.

Referee: "for notational consistency the Renyi index 'α' in Eqs. (72)-(73) should probably be replaced by 'n'"

The alpha in (72)-(73) is a real parameter, whereas the n in Section 3 is integer. Therefore we decided to keep the two symbols different. In the new version we used 'n' in Eqs (74) and (77).

Referee: "The paragraph after 'Property 4.2', which mentions the unbounded growth of operators on half-integer sites, could be expanded. This is an important result of this paper, so it should be discussed more. In particular, do the authors think the growth is linear? or sublinear? maybe logarithmic? This does not look obvious from Fig. 1."

See our above response to the referee's main point.

Referee: "The conclusion of Sec. 4.3 would probably be clearer if the final result (95) was given directly for the Renyi entropy (instead of the present form of Eq. (95), where the reader needs to recall what tr[(~B[a])2] is)."

We agree. Accordingly, in the revised version we rewrote the equation in terms of the Renyi entropy.

Referee: "In the conclusion, I don't quite understand the sentence 'for qubit chains, circuits with solitons (integrable or not) seem to be the only case where the operator complexity does not grow'. What about qubit chains that map to free fermions (XX chain, Ising chain, etc.)? Those have Hamiltonian evolution and the operator complexity is also known to remain constant for a subset of operators (see e.g. Ref. [22]). In what sense is that different, in terms of operator complexity?"

With that sentence we wanted to stress that here the complexity growth of all ultra-local operators is bounded, whereas in the examples mentioned by the referee some operators (namely the order parameters) have logarithmically growing complexity.

Referee: "some typos:
page 2: 'In Section 3 recall' -> 'In Section 3 we recall'
page 13: Eq. (83) is an empty line
page 14: 'originates a similarity' -> 'originates from a similarity'"

Fixed, thanks!

---

## Round 2 · List of Changes

1- Sentence about chiral circuits added to the abstract;

2- Definition of solitons updated;

3- Wedge replaced by "and" in Eqs. (5b) and (5c);

4- Discussion on integrable circuits improved;

5- Discussion after Eq. 13d expanded;

6- Discussion after Eq. 27 expanded;

7- Discussion after Property 4.2 expanded; Figure 2 and Table 1 added;

8- Various typos and errors in references corrected.

---

## Round 3 · Referee Report · Anonymous · 2020-3-16

Report

All questions suitably answered. Paper accepted for publication.

---

## Round 3 · Author Response

Dear Editor,

We thank the referee for her/his careful reading of our manuscript, for her/his positive assessment.

Here is a response to her/his only query.

"1: A precise comparison between the eigenvalues/eigenvector properties of transfer matrices yielding respectively linear behavior in I and log or constant behavior in II."

In the revised version of Paper I we now discuss how the “completely chaotic” class is incompatible with conservation laws, exhibiting exponentially many “additional” (to the x+1 given in Paper I) eigenvectors corresponding to eigenvalue 1 of the horizontal and vertical transfer matrices that one can construct in the presence of conserved quantities. Here (in the revised Section 3) we now refer to that discussion.

---

## Round 3 · List of Changes

- Sentence added after Eq 60

- Typos fixed in Eq 89 and 95

---

## Editorial Decision

published